



# Ultraviolet Radiation modelling from ground based and satellite measurements at Reunion Island, Southern Tropics

Kévin Lamy[1], Thierry Portafaix[1], Colette Brogniez[2], Sophie Godin-Beekman[3], Hassan Bencherif[1,4], Béatrice Morel[5], Andrea Pazmino[3], Jean Marc Metzger[6], Frédérique Auriol[2], Christine Deroo[2], Valentin Duflot[1], Philippe Goloub[2], and Charles N. Long[7]

[1]LACy, Laboratoire de l'Atmosphère et des Cyclones (UMR 8105 CNRS, Université de La Réunion, Météo-France), Saint-Denis de La Réunion, France
[2]Laboratoire d'Optique Atmosphèrique (LOA), Université des Sciences et Technologies de Lille, Lille, France
[3]Laboratoire Atmosphères, Milieux, Observations Spatiales, Service d'Aéronomie (LATMOS), CNRS, Institut Pierre Simon Laplace, Pierre et Marie Curie University, Paris, France
[4]School of Chemistry and Physics, University of KwaZulu Natal, Durban, South Africa
[5]Laboratoire d'Energétique, d'Electronique et Procédés (LE2P) , Université de la Réunion, France
[6]Observatoire des Sciences de l'Univers Réunion (OSU-R), UMS 3365 , Université de la Réunion, Saint Denis de la Réunion, France
[7]National Oceanographic and Atmospheric Administration, Earth System Research Laboratory, Boulder, Colorado, USA

*Correspondence to:* K. Lamy (kevin.lamy@univ-reunion.fr)

**Abstract.**

Surface ultraviolet radiation (SUR) is not an increasing concern after the implementation of the Montreal Protocol and the recovery of the ozone layer (Morgenstern et al., 2008). However large uncertainties remain in the prediction of the future changes of SUR (Bais et al., 2015). It has long been known that its variability depends on ozone levels and also on clouds, aerosol and albedo. It is therefore essential to monitor its evolution in the context of global change. Unfortunately there are few ground-based stations measuring surface UV irradiance in the southern tropics and particularly in the Indian Ocean, and long-term time series are required to study SUR variability and its relationship to ozone and to detect trends.

Reunion Island is located in the tropics (21S, 55E), in a part of the world where the amount of ozone in the ozone column is naturally low. In addition, this island is mountainous and the marine atmosphere is often clean with low aerosol concentrations. Thus, measurements show much higher SUR than at other sites at the same latitude or at mid-latitudes. Several studies pointed out that UV-B impacts the biosphere (Erickson III et al., 2015), especially aquatic system, which plays a central part in biogeochemical cycle (Hader et al., 2007). It can affect phytoplankton productivity (Smith and Cullen, 1995). This influence can result in either positive or negative feedback on climate (Zepp et al., 2007). In order to quantify the future evolution of SUR in the tropics, it is necessary to validate a model against present observations. This study is designed to be a preliminary parametric and sensitivity study of SUR modelling in the tropics.

Ground-based measurements of SUR have been performed at Reunion Island by a BENTHAM DTMc300 spectroradiometer since 2009. This instrument is part of the Network for the Detection of Atmospheric Composition Change (NDACC). In this study, we used the UltraViolet Index in order to quantify SUR radiation integratively.





We developed a local parametrization using the Tropospheric Ultraviolet and Visible Model (TUV (Madronich, 1993)) and compared the output of TUV to multiple years of BENTHAM spectral measurements. This comparison started in early 2009 and continued to 2016.

Only clear sky ultraviolet radiations were modelled, so we needed to sort-out the clear sky measurements. We used two
methods to detect cloudy conditions, the first based on an observer's hourly report of the sky cover, the second applying Long and Ackerman's (2000) algorithm to broadband pyranometer data to obtain the cloud fraction and then discriminate clear sky windows on SUR measurements. Multiple model inputs were tested and the output was evaluated against clear sky UVI observations. For total column ozone, we used ground-based measurements from the SAOZ spectrometer and satellite measurements from the OMI and SBUV instruments, while ozone profiles came from radio-soundings and the MLS ozone
product. Aerosol optical properties came from a local aerosol climatology established using a CIMEL photometer. The model sensitivity was investigated with respect to clear-sky SUR measurements.

Long et al.'s (2006) algorithm, with the co-located pyranometer data, gave better results for clear sky filtering than the observer's report. Model sensitivity to different extraterrestrial spectra depended on the total ozone column and the solar zenith angle and varied from 2.7 to 3.5% on UVI, a similar dependency being observed for the ozone cross sections with a smaller
difference, around 2.6%. Since the mean difference between various inputs of total ozone column was small, the corresponding response on UVI modelling was also quite small, at about 1%. The Radiative Amplification Factor of Total Ozone Column on UVI was also compared for observations and model. Finally, we were able to estimate UVI at Reunion Island with, at best, a mean relative difference of about 0.5 %, compared to clear sky observations.

# 1   Introduction

Ozone recovery prevented an important increase of Surface Ultraviolet Radiation (SUR) level (Morgenstern et al., 2008). Large uncertainties remain in the prediction of the future changes of SUR (Bais et al., 2015). As overexposure to this radiation is the main factor for the development of non-melanoma and melanoma skin cancers. Non-melanoma skin cancer is induced by chronic exposure and melanoma is induced by repeated burning and chronic exposure (Matsumura and Ananthaswamy, 2004). Holick et al. (1980) studied the beneficial effect of UV radiation on health through the synthesis of pre-vitamin D and numerous
studies have assessed the balance between benefits and risks. McKenzie et al. (2009) looked at the relation between erythemal weighted UV (Mc Kinlay and Diffey, 1987) and vitamin-D weighted UV (MacLaughlin et al., 1982). This work showed that, during winter at mid-latitudes vitamin D can be produced in a few minutes while avoiding the skin damage that occurs after an hour of exposure, depending on the skin area exposed. Behavioural studies have also been conducted in order to understand human activities in relation to UV intensity. Outdoor sports activities without sufficient solar protection have been shown to
increase the risk of developing skin lesions in childhood (Mahé et al., 2011). Tourism in northern mid-latitude cities in summer can also present a non-negligible risk of skin cancer (Mahé et al., 2013) . UV exposure and sun-protective practices during childhood were also investigated in New Zealand (Wright et al., 2007), and differences in children's exposure were explained



by their different activities. Sunburn risks among children and outdoor workers were evaluated in Reunion Island and South Africa. High values of cumulative daily ambient solar UV radiation were found for the three sites studied (Wright et al., 2013).

Total solar irradiance at the top of the atmosphere is, of course, the source of surface UV radiation. Its intensity varies directly with the sun radiative intensity. The sun has an 11 year solar cycle period (Willson and Hudson, 1991) which has a

direct influence on total solar irradiance at the top of the atmosphere. Total solar irradiance at the top of the atmosphere is also modulated by the variation in Earth's orbital parameters, which should be taken into account in very long-term climate studies but can be neglected in multi-decadal studies such as ours.

SUR is attenuated by the atmosphere and scattering processes from the top of the atmosphere to the surface. Investigating SUR variability from one year of ground-based measurements, McKenzie et al. (1991) showed that the dominant variation

of SUR was linked to SZA, while attenuation by clouds could exceed 50% and a total ozone column reduction of 1% could induce an increase of SUR of $1.25 \pm 0.20\%$.

Due to the depletion of the ozone layer by human made halogenated substances and the important impact of ozone on climate change, a major observing programme was set up to monitor atmospheric ozone in the last decades. The latest assessment of the state of the ozone layer (WMO, 2014) reported the end of the stratospheric ozone decline since the 1990s, with a stabilisation

of ozone levels at about 2% below those observed in the early 1980s. However, global circulation model simulations predict an acceleration of the Brewer Dobson circulation over the next century (Butchart, 2014), which would lead to a decrease of ozone levels in the tropics and an enhancement at higher latitudes (Hegglin and Shepherd, 2009). Clouds and Aerosols are also being intensively investigated, given their role in the climate energy budget and the fact that their radiative forcing remains the main uncertainty for climate studies (Boucher et al., 2013). They also are the main uncertainty factors in the future projections

of the solar UV irradiance (Bais et al., 2015). Global maps of UV-absorbing aerosols were derived by Herman et al. (1997). Later Krzyścin and Puchalski (1998) showed that a 10% decrease of aerosol optical thickness (AOT) at 550nm induced a $\sim$ 1.5% increase in UV erythemal daily dose. More recently Kazadzis et al. (2009) investigated the aerosol forcing efficiency in the UVA region, between 325nm and 340nm they found a mean reduction of irradiance of 15.2 %. It has been shown that clouds can reduce SUR variability ((Bais et al., 1993), (Calbó et al., 2005)) but broken clouds can also enhance it under specific

conditions (Mayer et al., 1998).

Those three time-evolving parameters being an important source of variability of UV radiation, we need to better understand their effects on surface UV. Radiative transfer modelling plays a key part in deducing UV evolution over the next century, but it needs to take account of a fair projection of those parameters. Climate models associated with radiative transfer models or empirical methods have been used to assess SUR evolution over the next century. Bais et al. (2011) and McKenzie et al. (2011),

respectively, found a 12% and 20% decrease of UVI at high latitude, a 3% to 5% decrease at mid-latitude and a 1% to 3% increase in the tropics. These projections depend strongly on the evolution of future climate, and Butler et al. (2016) presented the complexities associated with future ozone change and therefore surface UV change.

The Ultraviolet Index (UVI) (WHO, 2002) is one of the most common parameters used to investigate the impact of SUR on human health. It is the weighted integration of the ultraviolet irradiance between 280 nm and 400 nm, with the weight depending

on the human skin's response to erythema Mc Kinlay and Diffey (1987). UVI modelling was investigated thoroughly by Badosa



et al. (2007), who tested multiple inputs and compared the results to observations at four different sites (Lauder (45.04° S), Boulder (40.01° N), Mauna Loa (19.53° N) and Melbourne (37.63° S)). They found mean relative differences in UVI between model and observations ranging from 10% to less than 0.1%.

Following these studies, the present work tries to improve the understanding of surface UV variability in the southern tropical region, a sensitive area where very few studies have been conducted. This article is intended to improve surface UV modelling by analyzing the model sensitivity to different inputs. Six years of ground surface UV measurements made with a BENTHAM DTMc300 spectroradiometer are analysed versus ozone, cloud, and aerosol data derived from ground and satellite measurements spanning the same time period. As discussed previously, climate model simulations predict a decrease of ozone levels in the tropics and their enhancement at higher latitudes. This study is designed to establish a fine parametrization of UVI modelling in the tropics in order to later couple radiative transfer modelling and a chemistry climate model to obtain precise UVI projection.

Reunion Island is a small tropical island located in the Indian Ocean at a latitude of 20.90° South and a longitude of 55.50° East. This island is very mountainous with a peak at 3070 m asl and a mean altitude of $\sim$ 600 m asl. Almost all of the ground measurements were made at Reunion Island University, which is situated in the north of the island at an altitude of 80.0 m and at less than 2 km from the coast. The atmosphere in the boundary layer is dominated by the ocean and is often clean with low aerosol concentration. The usual weather follows a typical pattern during the day, due to the trade winds: the sky is usually free of clouds at sunrise but, clouds start to appear during the day as the trade winds blow on to the mountain. There is a strong contrast between the east and west sides of the island, the accumulation of clouds leading to more precipitation on the east coast while the west coast is mostly dry.

Surface UV measurements show a very high UV index compared to other sites at the same latitude (Lee-Taylor et al., 2010). Aerosol optical thickness shows a mean value of $\sim$ 0.07 at 440 nm, with the occasional maximum at $\sim$ 0.3. These multiple conditions : mountainous, tropical island with low aerosol concentrations, makes Reunion Island an interesting site for studying surface UV radiation. Since the island presents a very high UVI, and the population lives at low to relatively high altitudes, the highest cities being located at about 1.5 km asl, surface UV radiation is an important health concern. Note that the time zone of the site is UTC+4:00.

In section 2, we will first present the different data and the radiative transfer model used in this study. As discussed previously, clouds are an important factor of UV variability and, since clouds are not well resolved in the radiative transfer model, we chose to work only on clear-sky UVI modelling here. The datasets used in this study will also be presented.

In section 3, we will address the filtering method used to select only UVI observations for clear-sky conditions with the use of broadband pyranometer global and diffuse solar irradiances and Long et al. (2006) algorithm. A brief comparison will be made with human observer's report of cloud sky coverage.

In section 4, we will discuss the radiative transfer model sensitivity to various input parameter. Multiple modelling cases were run using different ozone and aerosol data at the time of the UV measurements, and also different input ozone cross sections and solar irradiance spectra at the top of the atmosphere. The impact on the UVI modelled will be analysed.



In section 5, the model will be validated against the clear-sky UVI observations at Reunion Island. We will investigate the model's ability to reproduce diurnal and seasonal variation of UVI, and its ability to reproduce the effect of total ozone column (TO3) variation on UVI. Lastly we will discuss the results and draw some conclusions.

## 2 Datasets

The multiple types of measurements used in this study, as input for the radiative transfer model or as reference to validate the model output, are summarised in Table 1.

UVI measurements were performed with a BENTHAM DTMc300 spectroradiometer. This spectroradiometer is composed of two monochromators and scans the wavelength range of 280-450 nm. According to Brogniez et al. (2016), Bentham DTMc300 UVI measurements have an expected uncertainty of about 5%. In this study, we used the standard erythemal ac-
10 tion spectrum published by the International Commission on Illumination to calculate erythemal weighted UV (CIE Standard, 1998). Simulation experiments under clear-sky conditions were conducted over the time period covered by SUR measurements, i.e. from 2009 to 2015.

TO3 measurements included:

- Ground based measurements from the SAOZ (Système d'Analyse par Observation Zenithal) (Pommereau and Goutail,
1988) UV-Visible spectrometer collocated with the Bentham.

- Satellite measurements from OMI (OMTO3 product) and SBUV (Bhartia and Wellemeyer, 2002).

And two different extraterrestrial spectra (ETS) measurements:

- Chance and Kurucz (2010)

- Dobber et al. (2008)

The impact of ozone cross sections (O3XS) absorption on UVI modelling was also investigated, O3XS was obtained from:

- Malicet et al. (1995) and Brion et al. (1998) (BDM) works which are currently used for SBUV instruments.

  - 280-345 nm at 295K, 243K, 228K and 218K from Malicet et al. (1995).
  - 345-450 nm at 195K from Brion et al. (1998).

- Bass and Paur (1985) (BP) are used currently for OMTO3.

- Gorshelev et al. (2014) and Serdyuchenko et al. (2014) obtained high spectral resolution ozone absorption cross sections (called SER hereafter) from a combination of Fourier transform and echelle spectrometers. Measurements, data analysis and comparisons are presented in the first paper, and temperature dependence is investigated throughout the second paper.



A joint initiative of the International Ozone Commission , the World Meteorological Organization and the Integrated Global Atmospheric Chemistry Observations has led to the Absorption Cross Sections of Ozone (ACSO) activity. The main objective of this activity is to evaluate the most suitable ozone absorption cross sections laboratory data to be used in atmospheric measurements. Orphal et al. (2016) published a recent status report on this activity. The conclusions are the following:

- BP data should no longer be used for retrieval of atmospheric ozone measurements.

- Ground-based measurements of total ozone and ozone profile should use SER,

- Both BDM and SER can be used for Ground-based Light Detection and Ranging (LIDAR) measurements.

- BDM should be used for satellite retrieval.

Ozone and temperature profiles came from radio-soundings, performed weekly at Reunion Island since 1993, and also
from MLS satellite measurements (Froidevaux et al., 2008). Nitrogen dioxide measurements were obtained with the SAOZ spectrometer. Aerosol measurements were derived from CIMEL sunphotometer measurements between 2009 and 2016. Cloud observer reports were made at Reunion Island every hour at about 10km from the UVI measurements, while global and diffuse total irradiance were measured every minute at the same location as the UVI measurements.

## 3   Clear-Sky Filtering

Clouds are known to play an important role in surface UV variability (Bais et al. (1993), Calbó et al. (2005) and Mayer et al. (1998)). As mentioned previously, this study was limited to clear sky UV observations and simulations. In order to filter out cloudy conditions, two different methods were used. The first, commonly used one was based on synoptic observer reports (SYNOP) made at a Météo-France weather station located about 10km from the UV measurements site. The observer reports follow WMO guidelines for cloud observation (http://worldweather.wmo.int/oktas). Sky observations are made every hour
and are quantified on a scale in oktas from 0 (clear sky) to 8 (totally overcast sky). We kept only UV measurements made for a cloudiness $\leq 1$ okta. Since the UV measurements are made every 15 minutes and sky observer reports are hourly, we interpolated these observations every 15 minutes. The effect of interpolation is taken into consideration and analysed below. The second method used Long and Ackerman (2000) and Long et al. (2006) algorithms. As input for this algorithm we used 1-min data of global and diffuse total irradiances measured at the same location as the UV measurements, with a SPN1 shaded
pyranometer. These algorithms performed multiple tests on the global, diffuse and direct irradiance in order to identify periods of clear skies. They have been validated against whole sky imager, lidar data and observer reports.

In order to compare the two methods, we tried multiple thresholds for the Cloud Observer Report (CF-SYNOP) and Cloud Fraction obtained with Long et al. (2006) algorithm, called CF-SWF hereafter. We considered that clear sky conditions prevailed when the cloud fraction was less than or equal to the CF-SYNOP or CF-SWF thresholds. From these we obtained UVI
filtered data, called respectively UV-SYNOP and UV-SWF hereafter.



We investigated numerous days and found CF-SWF to be more responsive and consistent with the UVI measurements. An example of a typical day with varying cloud fractions is represented in Figure 1. UVI corresponding to all sky conditions (UV-ALL) are marked in black circles, the blue circles represent UV-SWF and the red ones UV-SYNOP. CF-SWF is also represented by the blue dashed line. A clear-sky day would produce a UVI diurnal cycle resembling a Gaussian shaped function centred on

the solar noon, while moderate to high cloud fractions would generally reduce UVI. In some cases, broken cloud conditions may increase UVI by 20% higher than clear-sky conditions (Cede et al., 2002). Early in the day, at about 4:00 UTC, both UV-SYNOP and UV-SWF are absent, as no clear sky conditions are detected: cloud fraction at that time is quite high and impacts the UVI slightly. Around 9:00 we clearly see the impact of a rising CF on UV-ALL which rapidly decreases. UV-SWF is absent but CF-SYNOP still labels UV-ALL measurements (UV-SYNOP) as clear-sky while they are clearly not. This example, among

many others, made us decide to use only UV-SWF for the UVI clear-sky measurements in all further work.

We then investigated the daily and monthly density of UVI measurements. In Figure 2a, average UVI data distribution through the day is represented for UV-ALL(green bars) and UV-SWF (blue bars), along with the mean CF-SWF (black dashed line). We can see that UV-ALL is equally distributed through the day, since UVI measurements are made every day with at 15 minute intervals. In contrast, for clear-sky data UV-SWF, there are more measurements available during the morning than in

the afternoon, which is anti-correlated with the mean climatological CF-SWF. As clouds tend to form during the day, clear-sky UV measurements are less frequent in the afternoon. In Figure 2.b, representing UVI data distribution during the year, we see that UV-ALL are not equally distributed. There are less UV-ALL data for the first four months, especially during March and April, due to a few failures and technical maintenance of the BENTHAM spectroradiometer during the 6 years analysed here. We then see a seasonal variation of the availability of clear sky UVI measurements. During the austral summer, there is

an increasing mean cloud fraction and therefore fewer clear-sky measurements. Since the solar zenith angle and total ozone column also follow a diurnal and annual variability, respectively, the uneven clear sky UVI distribution through the day or through the year will induce a statistical bias on the following comparisons.

## 4   UV Modelling

### 4.1   Radiative Transfer Model

For UVI modelling, we used the Tropospheric Ultraviolet Visible (TUV) radiative transfer model version 5.3 (Madronich, 1993). TUV is available with two different radiative transfer schemes. We used the pseudo-spherical 8th-stream discrete ordinates (psndo.f) (Stamnes et al., 1988). The computation time is higher than with the generalised 2-stream method (Toon et al., 1989) also available in TUV but psndo.f is more accurate (Petropavlovskikh and Brasseur, 1995). Multiple parameters were modified in the model in order to reproduce the UVI measurements and site-specific climatology.

– Extraterrestrial Spectrum (ETS)

          – Solar Zenith Angle (SZA)

          – Total Ozone Column Amount (TO3)

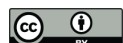



- Total Nitrogen Dioxide Column (TNO2)

- Ozone Profile (O3P)

- Temperature Profile (TP)

- Aerosol Optical Thickness (AOT) at 340 nm

- Aerosol Ångström exponent ($\alpha$) at 340-440 nm

- Single Scattering Albedo (SSA) ((Takemura et al., 2002) and (Lacagnina et al., 2015))

- Ground Surface Albedo (alb)

- Altitude (z)

Due to the lack of reliable data, total column sulphur-dioxide (TSO2) was set to zero, which could induce a modelling error

during a volcanic eruption. Between 2009 and 2015, there were a few volcanic eruptions of the Piton de la Fournaise, which is located on the opposite coast of Reunion Island to the site where the UVI measurements were taken. Unfortunately there is no TSO2 data available for this period. Following McPeters and Labow (2012), a monthly climatology of ozone and temperature profile was derived from ozone soundings and MLS satellite measurements. Single Scattering Albedo from the CIMEL sunphotometer was not usable as Dubovik et al. (2000) showed that SSA has an uncertainty higher than 100% if the AOT is

lower than 0.3, which was almost always the case here. As proposed by to Lacagnina et al. (2015) and Takemura et al. (2002), a fixed SSA of 0.95 was set.

Following Koelemeijer et al. (2003), surface albedo was taken to be constant at 0.08. According to Koepke et al. (1998), the UVI modelling error is about 5 % if the sum of the different uncertainties in input is considered at about 5%.

### 4.2   Influence of input parameters

To study the impact of various inputs on surface UV calculations, we tested multiple inputs from a baseline configuration (Table 2). Then different cases, called RTUV for Reunion Tropospheric UltraViolet hereafter, were run, with only one parameter varying in each (Table 3).

### 4.2.1   Solar Position and Extra-terrestrial Spectrum

In order to take into account the varying earth-sun distance (esd, in Astronomical Unit), a time dependent coefficient (ESF) is

used in TUV: $ESF(t) = 1/ESD^2$. The new extraterrestrial spectrum ET' , which is the spectrum at the top of the atmosphere corrected for any instant of time (it) is then (1):

$$ET' = ESF(it) * ET \tag{1}$$

For convenience, we will now refer to the corrected extraterrestrial spectrum simply as ET. The ESD correction is done when RTUV is used for time series studies but, in order to study the sensitivity to ET spectrum (or to the O3XS later in 4.2.2) we





chose to run TUV in an idealized state (i-RTUV), with constant earth-sun distance, total ozone column varying from 250 DU to 350 DU, solar zenith angle varying from 0° to 60°, and mean values of aerosol representative of the entire study period (see Table 3).

Since we want to understand the sensitivity to ET spectrum, i-RTUV was run with Dobber et al. (2008) or with Chance and Kurucz (2010) ET spectrum. From there we took UVI output and defined the UVI relative difference (RD) between these two cases as :

$$RD[\%] = 200 * \frac{UVI_{dobber} - UVI_{chance}}{UVI_{dobber} + UVI_{chance}} \qquad (2)$$

RD on the surface UVI modelled between the two ET spectra for the SZA and TO3 studied is represented on Figure 4a for a monthly climatological OP and TP of January and on Figure 4b for October. The highest RD appears to occur for low SZA and low TO3, at about 5% for January and about 4% for October. Minima of RD are for high SZA and high TO3, at about 3% and 2% for January and October, respectively. At fixed TO3, when SZA increases, the path length travelled by the radiation crossing the atmosphere traveled is longer and other processes, such as Rayleigh scattering, have more impact on the surface UVI modelled than different ET spectra do. At fixed SZA, when TO3 increases, the RD decreases, or when ozone molecular absorption increases the differences in the ET spectra are less important.

During the month of October, OP shifts from the annual mean (Figure 3). There is an increase in tropospheric ozone through the arrival of emissions due to biomass burning over the western part of the Indian Ocean zone Baldy et al. (1996). The absorption effectiveness of tropospheric ozone relative to stratospheric ozone depends on SZA (Brühl and Crutzen, 1989). This is why there is a small difference (about 1%) between January and October. This difference is only due to the shift in ozone and temperature distributions.

On Figure 4c, the monthly mean of UVI RD between RTUV05 and RTUV01 is represented by a dashed blue line. These two RTUV cases have the same configuration except for the ET spectrum. The sign of the RD here is coherent with equation (2), i.e. RTUV05 (Chance ET) minus RTUV01 (Dobber ET). The monthly mean of RD between these cases oscillates between 2.7 and 3.4%, these values being of the same of order than as the i-RTUV cases. The oscillation observed here is anti-correlated with the maximum of SZA at solar noon at the site studied.

To conclude, UVI dependency on the ET spectra is higher at low SZA and low TO3 but these are also the conditions where we find the maximum of UVI, i.e where the health risk is highest. While the RD is dependent on SZA and TO3 (at about 3.0%), OP and TP also constitute a factor of variability (about 1%). RD is modulated through the year with the displacement of SZA. The Chance's spectrum impact on the surface UVI modelled will be an increase of about 3.0% with respect to Dobber's.

### 4.2.2 Ozone Cross Sections

Following these studies we investigated the impact on simulated UV irradiance modelled with different absorption cross sections of ozone (O3XS) given by the previously cited, BDM and SER. We define the relative difference between the two modelling cases as :

$$RD = 200 * \frac{x_{BDM} - x_{SER}}{x_{BDM} + x_{SER}} \qquad (3)$$




Figure 5a represents the UVI RD between two different O3XS (equation 3) in an idealized run (i-RTUV). Here, RD ranges from 2.40% to 2.9%. We observe the same variability as for the ET sensitivity: highest values of RD correspond to low SZA and low TO3 and as either SZA or TO3 increases, UVI RD will decrease. Figure 5c shows the corresponding RTUV cases (RTUV06 with BDM O3XS and RTUV01 with SER O3XS), which differ only by O3XS. There is also a yearly UVI RD oscillation anti-correlated with the maxima of SZA at solar noon. The mean RD is about 2.7% which is consistent with the i-RTUV case. The oscillation around the annual mean is smaller here than for the ET spectra (about 0.5%) but so is the mean RD.

To conclude on O3XS for this study site, using SER O3XS instead of BDM O3XS increases the mean surface UVI by about 2.7%. This RD is modulated by the SZA, TO3, OP and TP.

### 4.2.3    Total Ozone Column

Three different TO3 datasets, covering the period 2009 to 2016 in RTUV01 (SAOZ), RTUV02 (OMTO3) and RTUV03 (SBUV) were investigated. In contrast to the simulations in the idealized state, these simulations were conducted with varying earth-sun distance. The Relative Difference distribution for TO3 and UVI are represented on Figure 6 with one percent bins. Mean RD between TO3 datasets is smaller than 0.3% with a standard deviation of about 2%. The impact on the surface UVI modelled is slightly higher. There is a small mean relative difference of $0.08 \pm 2.31\%$ between SBUV and OMTO3, $0.36 \pm 2.0\%$ between SAOZ and OMTO3 and a $-0.31 \pm 1.86\%$ between SAOZ and SBUV.

Since TO3 and surface UVI are anti-correlated, a positive RD in TO3 will lead to a negative RD on UVI and vice versa. This is expressed by the opposite sign of the mean RD between TO3 and UVI, and also by the RD distribution. If the distribution of RD of TO3 tends to shift towards positive values, the distribution of UVI RD will shift towards negative values.

### 4.2.4    Aerosol Climatology and Observations

For the aerosol input, two datasets were used, one a monthly climatology derived from the CIMEL sunphotometer measurements from 2009 to 2016 (Cimel Clim), and the other the daily mean from CIMEL measurements (Cimel Daily). In order to understand the importance of aerosol measurements used in surface UVI modelling, we define the Aerosol Daily Anomaly (ADA) as the difference between AOT monthly climatology and the daily mean:

$$ADA = Aerosol_{clim} - Aerosol_{daily} \tag{4}$$

Two RTUV cases are identical except for the aerosol inputs: RTUV04 was run with Cimel Daily and RTUV 01 with Cimel Clim.

Figure 7a represents the ADA for the entire study period. Aerosol optical thicknesses are usually quite low but can rise quickly. We can observe a strong peak of 0.25 AOT anomaly and also a decrease to about 0.1 AOT anomaly. The maximum anomaly usually appears at the end of the year. This corresponds to the arrival of biomass burning emissions over the western part of the Indian Ocean zone (Baldy et al., 1996). Figure 7b shows UVI RD between RTUV04 and RTUV01. This succession





of AOT anomalies has a direct impact on surface UVI, for the 0.25 increase in AOT during the end of 2010, there is a ∼30% relative difference between the two surface UVI modelled.

We can clearly observe the impact of anomalous peaks of AOT on climatological AOT in Figure 7c, for wavelengths in the ultraviolet spectrum (340, 380 and 440 nm). There is a strong peak centred around October. This monthly distribution of high

AOT anomalies and high relative difference on modelled UVI is confirmed on Fig.7d and Fig.7e, which represent the monthly distribution of AOT anomalies or UVI relative difference split into 3 categories. Quantitatively more AOT anomalies (more than 0.15 ADA) are centred around October, which leads to higher values of UVI relative difference. On Figure 7f, the monthly climatologies of SSA (at 438 nm, 669 nm, 871 nm and 1022 nm) derived CIMEL measurements are represented. These values are very uncertain, as presented by Dubovik et al. (2000), SSA can not be determined correctly by the CIMEL sunphotometer

if AOTs are lower than 0.3. On Figure 7g, the Ångström exponent is represented for five pairs of wavelengths. The Ångström exponent describes the spectral dependence of the AOT; it is directly affected by the increasing AOT in October.

To conclude on aerosols, even though the mean relative differences in surface UVI for the two cases are very low for the entire study period (-0.40%), there is still a punctual effect where surface UVI could be overestimated by ∼30%.

## 5   Model Validation

In this section, we compare observations made only in clear-sky conditions

### 5.1   Radiative Amplification Factor

In order to study the sensitivity of the modelling output, we need to understand the variability of UVI and ozone. The scaling function between UVI and ozone is commonly described as the Radiative Amplification Factor (RAF). In Booth and Madronich (1994), two methods for retrieving RAF are presented :

- The first one is based on a linear relationship between UVI and TO3. We proceed by considering every total ozone column measurement at a specific date t and the corresponding UVI. We then compute the relative difference, for a specific SZA interval, between all pairs of UVI and TO3, such as at two dates t1 and t2 :

$$\frac{\Delta UVI}{UVI_{t_1}} = \frac{UVI_{t_1} - UVI_{t_2}}{UVI_{t_1}} \tag{5}$$

$$\frac{\Delta TO3}{TO3_{t_1}} = \frac{TO3_{t_1} - TO3_{t_2}}{TO3_{t_1}} \tag{6}$$

This is also done for UVI obtained from the RTUV01 modelling case (see Table 3). Following Booth and Madronich (1994), we then define the linear Radiative Amplification Factor as:

$$RAF_L = -\frac{\Delta UVI}{UVI_{t_1}} \Big/ \frac{\Delta TO3}{TO3_{t_1}} \tag{7}$$





- By following Madronich (1993), a second method is used to obtain the RAF. It is based on a power law relationship:

$$\frac{UVI_{t_1}}{UVI_{t_2}} = \left(\frac{TO3_{t_1}}{TO3_{t_2}}\right)^{-RAF_P} \tag{8}$$

$$RAF_P = \ln\left(\frac{UVI_{t_1}}{UVI_{t_2}}\right) \Big/ \ln\left(\frac{TO3_{t_2}}{TO3_{t_1}}\right) \tag{9}$$

In previous works, Booth and Madronich (1994) found a $RAF_L$ of 1.1 from broadband measurements in Antarctica. More

recently, from a theoretical point of view, Herman (2010) found a $RAF_P$ of 1.25 at low SZA, decreasing to 1.1 at higher SZA. These values correspond to a mid-latitude profile. The higher RAF values found here can be explained by the very clear sky, the low latitude (20.90° S), and the lower value of TO3. Bodhaine et al. (1997) analysed one year of UV measurements at Mauna Loa, Hawaii, and found $RAF_P$ values between about 1.3 at a SZA of 15° and 0.6 at 85° SZA. At a SZA of 45°, while Bodhaine et al. (1997) found a $RAF_P$ of $1.38 \pm 0.2$, McKenzie et al. (1991) found a $RAF_P$ of $1.25 \pm 0.20$ at Lauder, New

Zealand (45° S).

Here, both RAF (linear and power) are calculated for an ideal modelling case (i-RTUV), for the observations (UV-SWF) and for a real-condition modelling case (RTUV04). The first objective is to evaluate the RAF of TO3 on the UVI for the three cases, to see how they compare to each other, and to determine whether RTUV01 is close to the observations (UV-SWF) by being able to reproduce the RAF of TO3. The second objective is to compare the RAFs found here with those found previously

in other studies at other sites.

On Figure 8a, ΔUVI/UVI is plotted against ΔTO3/TO3 for a SZA of 45° $\pm 0.1°$ for the three cases. The best fitting curve of each case is obtained from a least squares fitting method, from which $RAF_L$ is also deduced (Booth and Madronich, 1994). These ranges of SZA are chosen due to availability of measurement. Because of annual variation of the SZA, low values of SZA occur during the rainy season, the season when we filter out most of our data. UV-SWF is in blue, RTUV01 in red and

20 i-RTUV in green. On Figure 8b, the same method is used in order to retrieve $RAF_P$ (Madronich, 1993). Figures 8c and 8d represent $RAF_L$ and $RAF_P$, respectively, for the three cases against SZA, with a two sigma dispersion bar.

For i-RTUV, $RAF_P$ (dashed green line) are at about 1.20 for a SZA of 25° decreasing to 1.14 for a SZA of 55°, $RAF_L$ (solid green line) decrease from 1.5 to 1.42. Between the same SZAs, while $RAF_P$ derived from the clear-sky observations (UV-SWF, dashed blue line) decreases from $2.00 \pm 0.35$ to $1.45 \pm 0.16$, $RAF_L$ (solid blue line) decreases from $1.88 \pm 0.35$

to $1.35 \pm 0.16$. From RTUV04, $RAF_P$ (dashed red line) ranged between $1.31 \pm 0.13$ and $1.19 \pm 0.06$. At a SZA of 45°, from the observations, the $RAF_P$ obtained was $1.39 \pm 0.12$, this value is close to that found by Bodhaine et al. (1997), of $1.38 \pm 0.2$, for a similar site in the tropics.

RAF tends to decrease as SZA increases. When the SZA increases the path travelled through the atmosphere will be longer and other processes, such as Rayleigh extinction, will have a higher impact on the UVI. Since the absorption effectiveness of

30 trospospheric ozone relative to stratospheric ozone depends on SZA (Brühl and Crutzen, 1989), the ozone distribution and the temperature profile have an impact on the RAF value. This is one of the reasons why both RAF deduced from i-RTUV cases present a smooth line. In i-RTUV, nothing changes except TO3 and SZA. While in RTUV01 and UV-SWF, there is a variation





in aerosols, TNO2, ESD, OP and TP between t1 and t2. Nonetheless $RAF_P$ and $RAF_L$ are close for UV-SWF and RTUV01 cases.

It is difficult to determine whether observations are better represented by a linear law or a power law as dispersion on the observations is high in both cases.

## 5.2 Validation against Observed Clear-Sky UVI

In order to validate UVI modelling for the southern tropics, we compared the output of multiple model cases against UVI clear-sky measurements. Table 4 presents relative difference and standard deviation for the six Reunion Tropospheric UltraViolet (RTUV) model cases. Closest agreement between measurements and model is found for the RTUV03 case. This corresponds to a configuration with daily aerosol measurements, Dobber ET spectrum and SER ozone cross sections. We compared measurements at SZA $\leq$ 60 degrees.

We define the Relative Difference between RTUV and UV-SWF as :

$$RD[\%] = 100 * \frac{UVI_{RTUV} - UVI_{OBS}}{UVI_{OBS}} \tag{10}$$

Results for the mean, standard deviation and median of the relative difference are presented in Table 4.

Of the three sets of ozone total column, RTUV03 (SBUV) is the best, with a mean relative difference (MRD) of $0.43 \pm 5.60$ %, 0.11% lower than RTUV02 (OMI) and 0.51% lower than the run with SBUV measurements as input. Standard deviations (STD) are about the same for the 5 cases, around 5.7%. RTUV03 obtains the lowest median (MED) at 0.44 %.

For different extraterrestrial spectra in RTUV01 (Dobber et al., 2008) and RTUV05 (Chance and Kurucz, 2010), a 3.24 % difference is found between both MRD and MED. The difference is consistent with the one found in Section 4.2.1.

The influence of the choice of ozone cross sections on UVI modelled can be analysed with RTUV01 (SER) and RTUV 06 (BDM). The first is 3.44% lower than the second. The difference between the two is higher here than the difference found previously (section 4.2.2) in an idealised run, where only one profile of temperature and one of ozone were used for the sensitivity test. BDM O3XS ((Brion et al., 1998) and (Malicet et al., 1995)) were calibrated for four temperatures, 295 K, 243 K, 228 K and 218 K, while SER was run with temperatures ranging from 193 K to 293 K with a 10K step. Since UVI is sensitive to the ozone and temperature profiles, different O3XS can induce higher differences in surface UVI than we calculated previously during the sensitive test. This needs to be further investigated, notably the impact of O3XS and ozone and temperature profiles on the surface UVI.

A linear regression representation and mean relative difference against SZA are given in Figure 9 for all RTUV cases. On all subplots of Figure 9a, data corresponding to a daily mean AOT higher than the monthly mean AOT are represented in red. All conditions are in blue. RTUV01 and RTUV04 use different aerosol inputs, RTUV01 was modelled with an aerosol daily mean and RTUV04 used a monthly climatology. The MRD of RTUV01 is lower than that of RTUV04, whereas the opposite is true for the MED. Unlike the mean, the median is insensitive to outliers. In section 4.2.4 it was noted that, even if MRD is very low, there are also singular peaks that can reach up to 30% MRD. These are the outliers that are taken into account in





RTUV04. Close examination of Figure 9a reveals consistent outliers under the fitted curves for RTUV01, 02, 03, 05 and 06, which correspond to an underestimation of the UVI (due to overestimation of AOT). For RTUV04, which is the case with daily mean AOT, there are only blue crosses. Bottom outliers on every other RTUV case can be explained by the use of an AOT monthly climatology which is not representative of the daily mean AOT. For RTUV04 there are a few overestimations of the UVI that do not appear on the other cases. This is probably due to the AOT variability during the day, which is not taken into account in the other cases.

The optimised input configuration is RTUV01. The corresponding statistical values are consistent with those found in other studies. Badosa et al. (2007) found a MRD lower than 10% concerning observations with a similar instrument and filtering techniques at Lauder (45.04° S), Boulder(40.01° N), Mauna Loa(19.53° N) and Melbourne(37.63° S). Mauna Loa can be compared with Reunion Island as both sites are in the tropics with similar weather conditions, MRD between Badosa et al. (2007) model cases and observations ranged from -1.8 % to 3.6 %. In the present study comparable values of MRD were found (ranging from 0.43 % to 4.38 %, which is well within the modelling uncertainty of 5 %). The filtering method at Mauna Loa was probably stronger due to the use of Sky Imager.

Diurnal cycles of RD are represented in Figure 10 for all the year and for two seasons, austral winter (June, July and August) and austral summer (December, January and February). RD tends to increase during the day for all year and both seasons due to the formation of clouds. Standard deviation is higher in the afternoon during summer, due to the strong presence of clouds in this period (see Figure 2). Since there are fewer clear-sky measurements for this period, the comparison is statistically weaker here, with increased the dispersion. In winter, there is more data available, and less filtering (see Figure 2), so standard deviations are smaller.

In Figure 11 the diurnal cycle of the UVI seasonal mean and maximum are represented. Consistent results are obtained. Mean non-filtered UVI are always lower than mean clear-sky values. As mentioned before, the model tends to overestimate clear-sky UVI. Both filtered data (UVI-SWF and UVI-SYNOP) are in agreement, UVI-SYNOP diverges from UVI-SWF during the afternoon for the austral summer but, as stated before, SYNOP observer reports are less accurate than SWF cloud fractions. This is probably due to the distance between SYNOP observer reports, which are about 10 km from SWF, and UVI measurements, and the sampling difference: 1h for SYNOP, 1 min for SWF and 15 min for UV measurements. It would be very interesting to add an all-sky camera on the same site for more accurate indications of cloudiness. Austral summer is usually a very cloudy season in this southern tropical region. A maximum of UVI appears for unfiltered data (red dashed line) with strong values, usually up to 18, around 8:00 UTC (local noon time). This is probably due to UV enhancement by cloud fractional sky cover. This phenomenon has been described before, for example by Calbó et al. (2005) and Jégou et al. (2011) and its quantification in the southern tropics will be the subject of a future study.

# 6 Conclusions

The physics of radiative transfer is a well understood but the modelling of surface UV radiation is still a challenge since multiple parameters need to be taken into account. For clouds, we can simply filter observations and work in clear-sky conditions.





We investigated the sensitivity of UV radiation modelling to various input parameters. The impact of different extraterrestrial spectra or ozone cross sections has not been investigated previously. For the ET spectrum, we found a relative difference between 2.7% and 3.5 % depending on the total ozone column and solar zenith angle. For the ozone cross sections, the relative difference ranged from 2.4 to 2.9%, also with a dependency on total ozone column and solar zenith angle. The impact was

higher for low SZA and low TO3 during the diurnal and seasonal maximum of UVI, i.e. when the burning efficiency of the radiation on human skin is higher. This difference was found to be dependent on the OP and TP.

For total ozone column and aerosols, the results were close to those of other studies carried out at different latitudes (Badosa et al., 2007). Badosa et al. (2007) also investigated the impact of ozone total column and AOT for four different sites: Lauder, New Zealand (45.04° S, 169.68° E), Boulder, Colorado (40.01° N, 105.25° W), Mauna Loa, Hawaii (19.53° N, 155.58° W)

and Melbourne, Australia (37.69° S, 144.95° E). They found that different sources of input for total ozone column (SBUV or TOMS satellite measurements or Dobson ground measurements) had an impact of $\sim$ 2% to 5% on the surface UVI modelled. Here we found monthly mean differences between -2.5 % and 7.5%. The mean difference of UVI sensitivity to the total ozone column was smaller than 0.4% between any two datasets. UVI RTUV01 (SAOZ) was higher than RTUV02 (OMTO3). OMTO3 was on average the highest total ozone column, which led to the lowest modelled values of UVI. Brogniez et al. (2016) found

that OMI-UVI products were higher than local measurements of UVI in Reunion Island. OMI-UVI products are based on the OMTO3 product, but also take other OMI products (aerosols, surface albedo) into account (Krotkov et al., 2002). Here, using OMTO3 as the only input parameter to retrieve UVI did not produce a strong positive bias. Following the same study for aerosol (Badosa et al., 2007), it was found that AOT was very low at Mauna Loa, with values centred around 0.08, and the ratio between UVI modelled with and without aerosol was always between 0.96 and 1.02. Here, we found small mean differences

between UVI modelled on daily measurements and climatological values of AOT (about -0.40%) but strong peaks in AOT could be missed and would yield UVI overestimations of $\sim$ 30%. Aerosol climatology could be used for climatic studies of surface UVI but should be avoided for short-term predictions and preventive action for the population concerned, especially during the biomass burning season.

We also investigated the relationship between total ozone column and UVI variations through the Radiative Amplification

Factor (from a linear and a power law). At a SZA of 29° for the observations (UVI-SWF), model (RTUV01) and idealized model (i-RTUV), a $RAF_L$ of respectively 1.83, 1.2 and 1.5 was found. At a higher SZA, 45° a lower $RAF_L$ was found: respectively 1.25, 1.01 and 1.47. $RAF_P$ was closer to $RAF_L$ for both the observations and the model but more consistently for the idealized model. It was higher than in other studies ((Booth and Madronich, 1994) (Herman, 2010)) but these were done at higher latitudes. The closest value found here to other studies was a $RAF_P$ of 1.39 $\pm$ 0.12 at a SZA of 45 °. Bodhaine et al.

(1997) found 1.38 $\pm$ 0.2 for a similar site in the tropics. In general RAF tends to decrease as SZA increases. The significant dispersions on the observations make it impossible to conclude on on the linear or power relation between Total Ozone Column and UVI.

As previously noted RAF tends to decrease as SZA increases, presumably because of various effects such as the influence of ozone, temperature profiles and Rayleigh scattering, which would reduce the impact of TO3 on UVI.





Clear-sky UVI in the southern tropics was modelled with a MRD of 0.43 ± 5.83 % up to 4.38 ± 5.78 %, which is within the modelling uncertainty of 4.5 %. MED values ranged from 0.44 % to 4.51 %.

Monthly climatology of filtered and unfiltered clear-sky conditions revealed few maximum values of UVI during all sky conditions, this phenomenon is due to multiple reflections on cloud edges in case of broken cloud cover.

5      Future study will be needed to to take this into account. TUV is a one-dimensional model but, to considerer backscattering we need to have at least two-dimensional radiative transfer modelling. Following these results, the next step will be a projection of UV changes in the southern tropics.



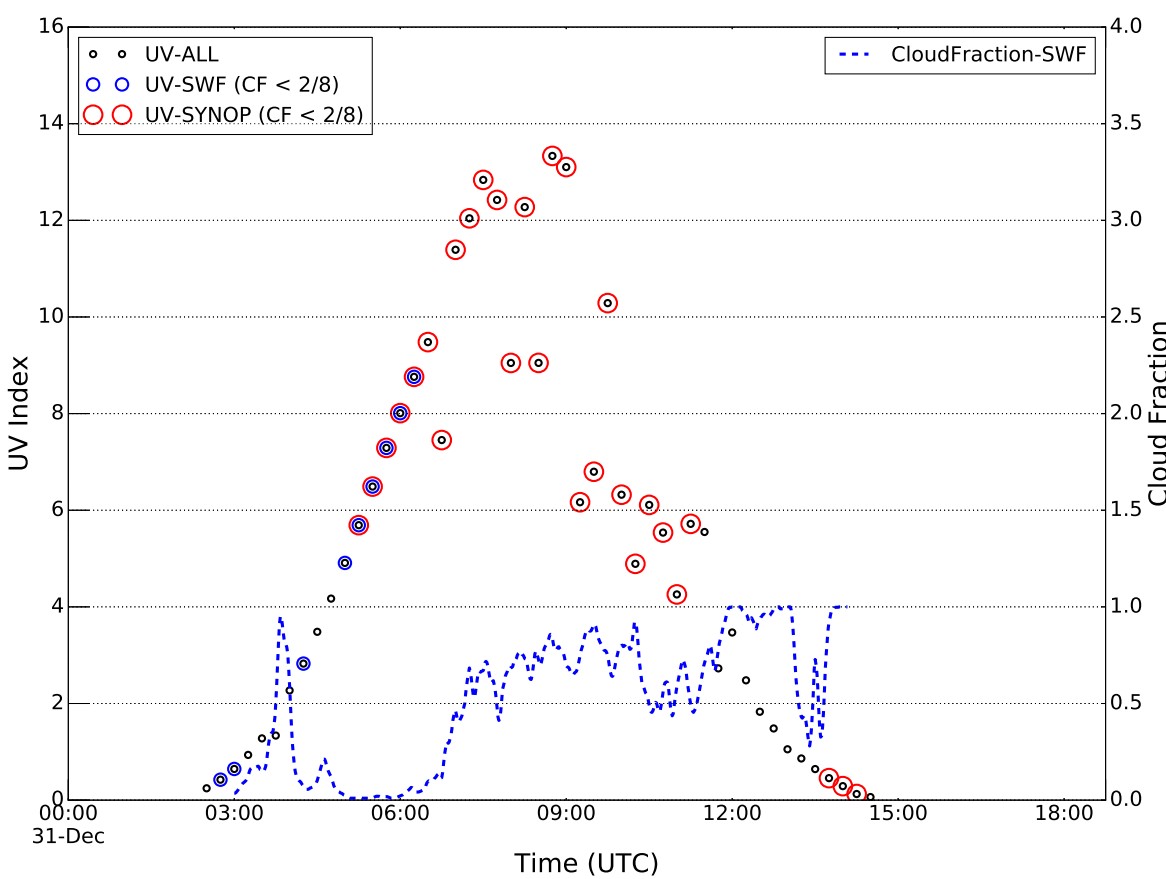

**Figure 1.** Daily UV index at La Réunion on December 31, 2010; for UV-all Bentham measurements in black circles, UV-SWF: UV data filtered with CF-SWF in blue circles, UV-SYNOP: UV data filtered with CF-SYNOP in red circles. Cloud Fraction (CF-SWF) is also represented by the blue dashed line.





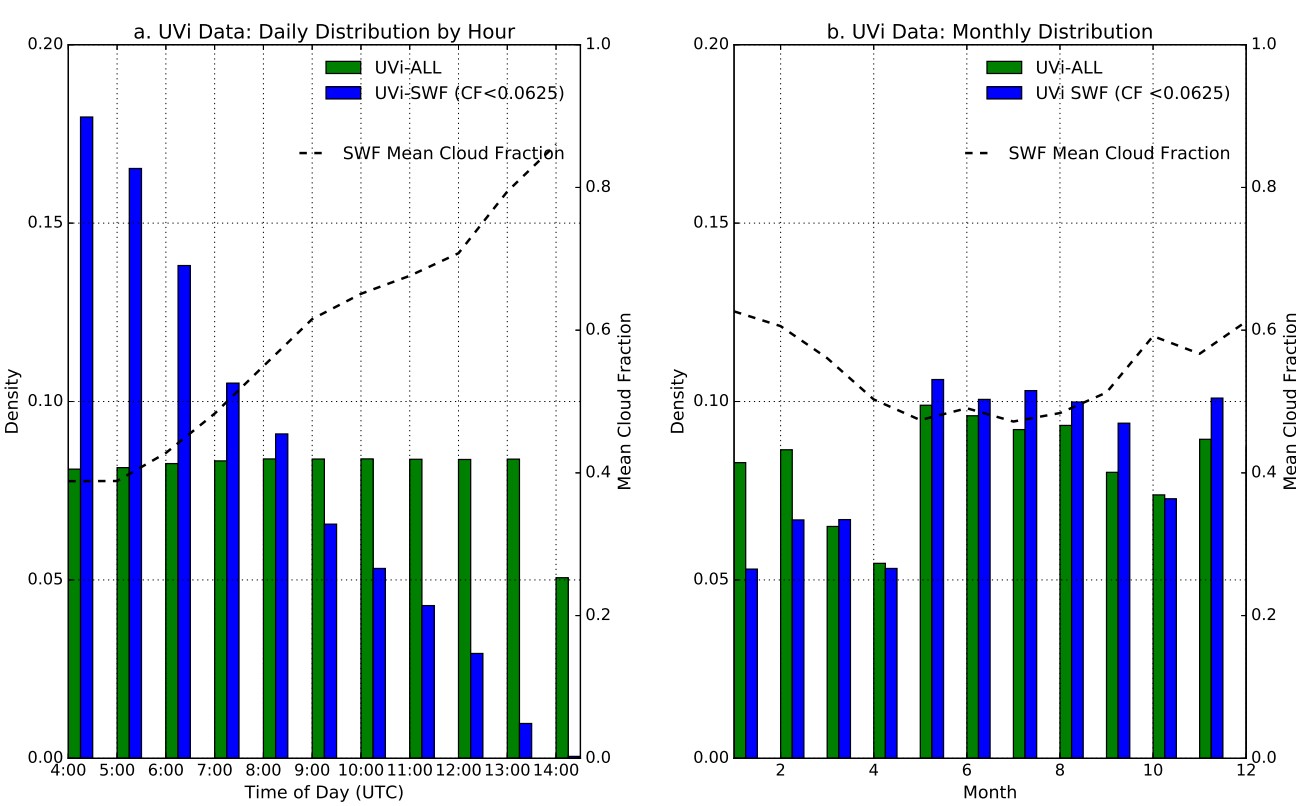

**Figure 2.** UVI data distribution and Cloud Fraction for 1h bins. UVI-all in green, UVI-SWF in blue, Cloud Fraction in dotted line.



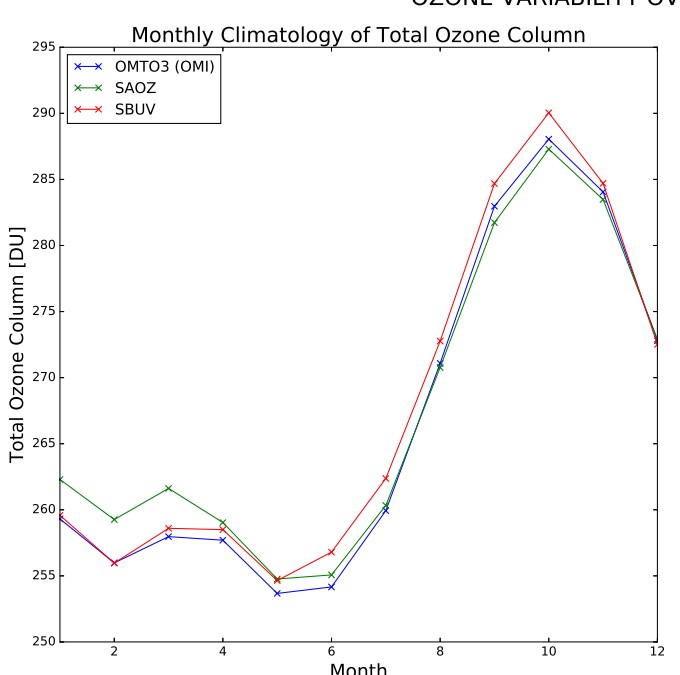
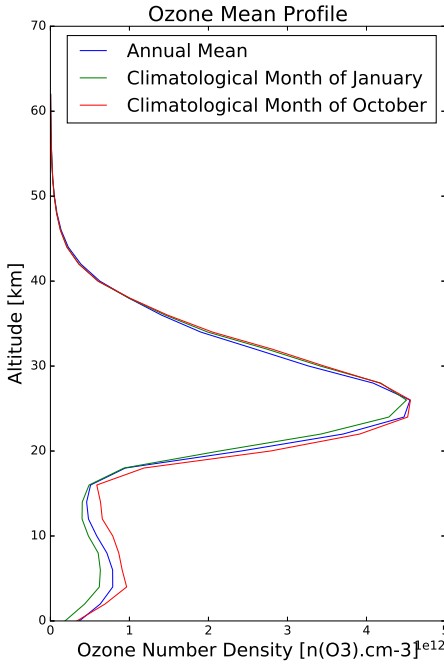

**Figure 3.** Ozone Variability over Reunion Island. Left : Monthly Climatology of Total Ozone Column for three datasets (OMI in blue, SAOZ in green and SBUV in red) Right: Mean Ozone Profile from McPeters and Labow [2012] Climatology. Annual mean in blue, January in green and October in red.





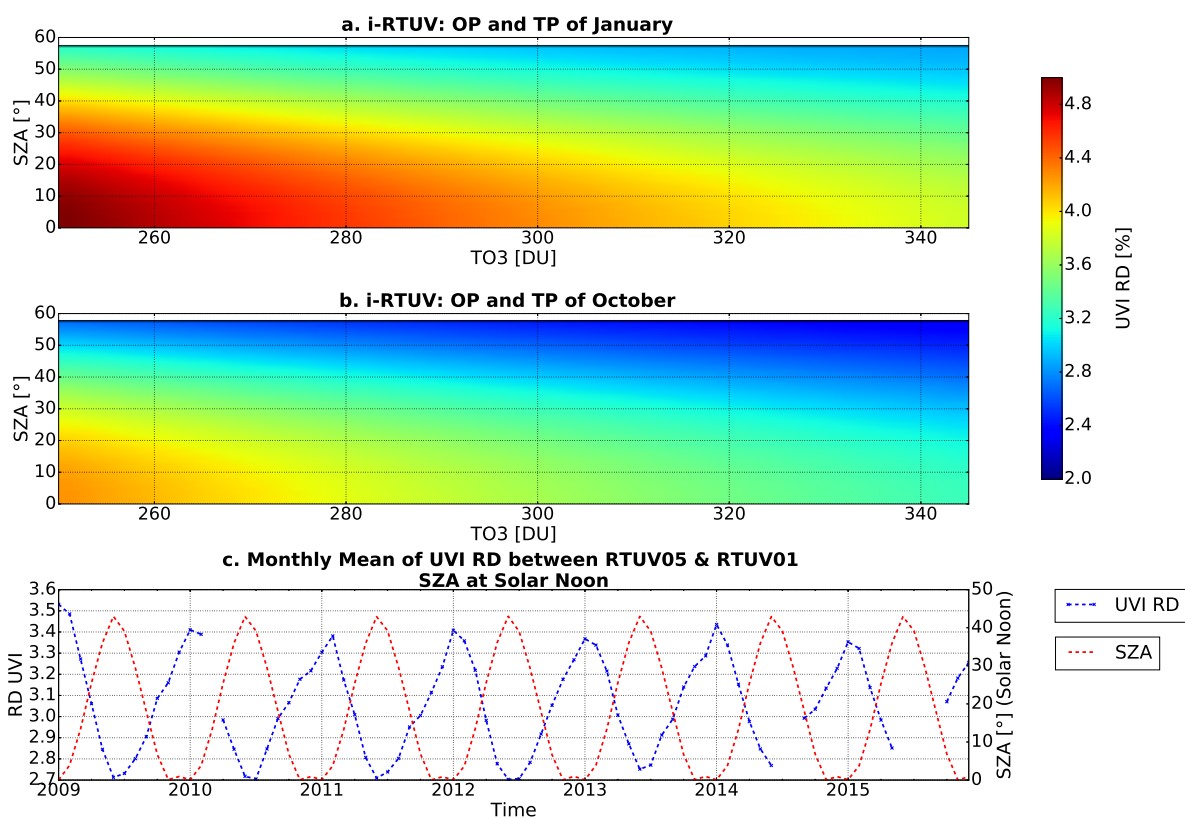

**Figure 4.** a) UVI Relative Difference between two idealized runs with different ET spectra for a varying TO3 and SZA for a climatological OP and TP of January b) Same as a) but for OP and TP of October. c) Monthly Mean of UVI RD between RTUV05 and RTUV01 (blue dashed line). Monthly Mean of SZA at Solar Noon. (red dashed line)



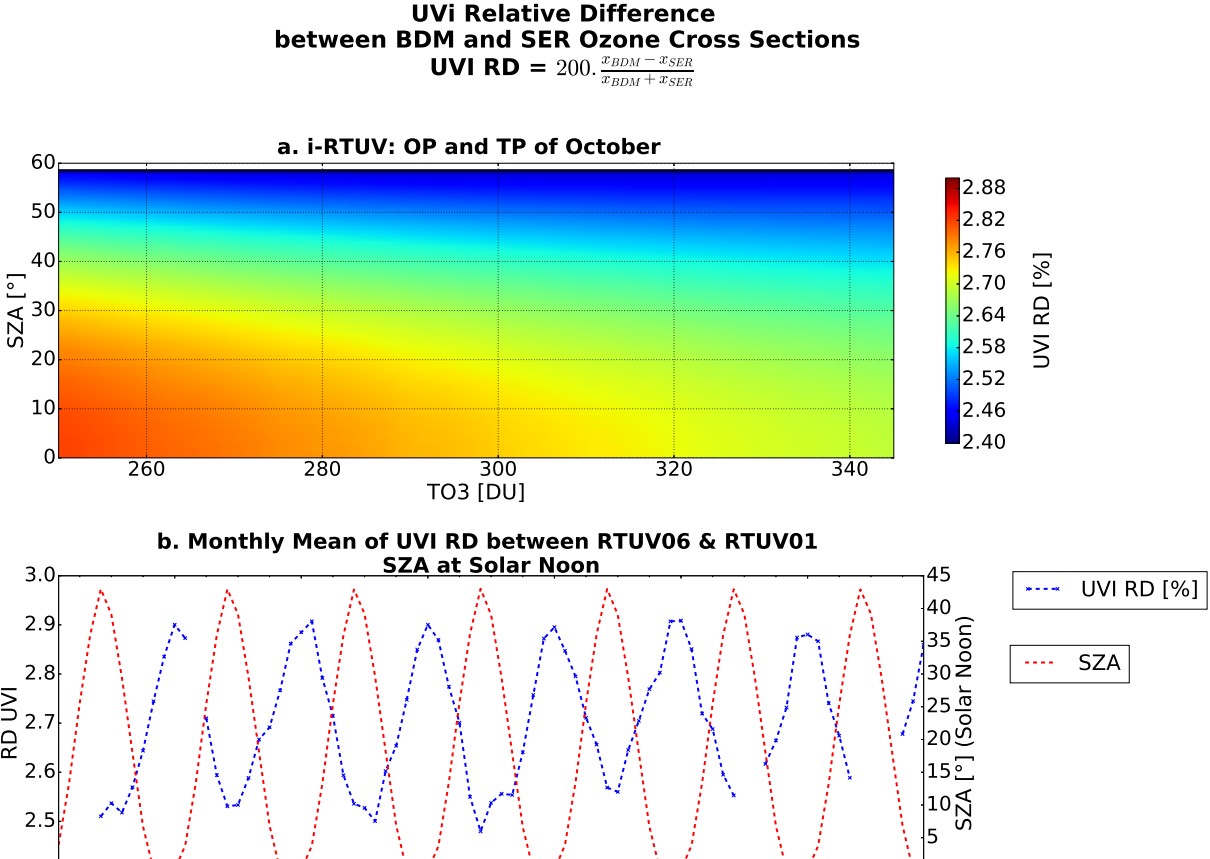

**Figure 5.** a) UVI Relative Difference between two idealizeds run with different O3XS for a varying TO3 and SZA and for climatological OP and TP of October. b) Monthly Mean of UVI RD between RTUV06 and RTUV01 (blue dashed line). Monthly Mean of SZA at Solar Noon. (red dashed line)





**Figure 6.** Distribution of Relative Difference [%] between different TO3 data sets and on the corresponding surface UVI modelled at all SZA.







**Figure 7.** Impact of aerosols on UVI modelling.

a) Aerosol optical thickness Anomalies during the study period. b) UVI Relative Difference between a run with Climatological Aerosols and Daily aerosols. c) AOT monthly climatology. d) AOT anomalies distribution. e) UVI Relative Difference monthly distribution. f,g) Single Scattering Albedo and Ångström Exponent Monthly Climatology.




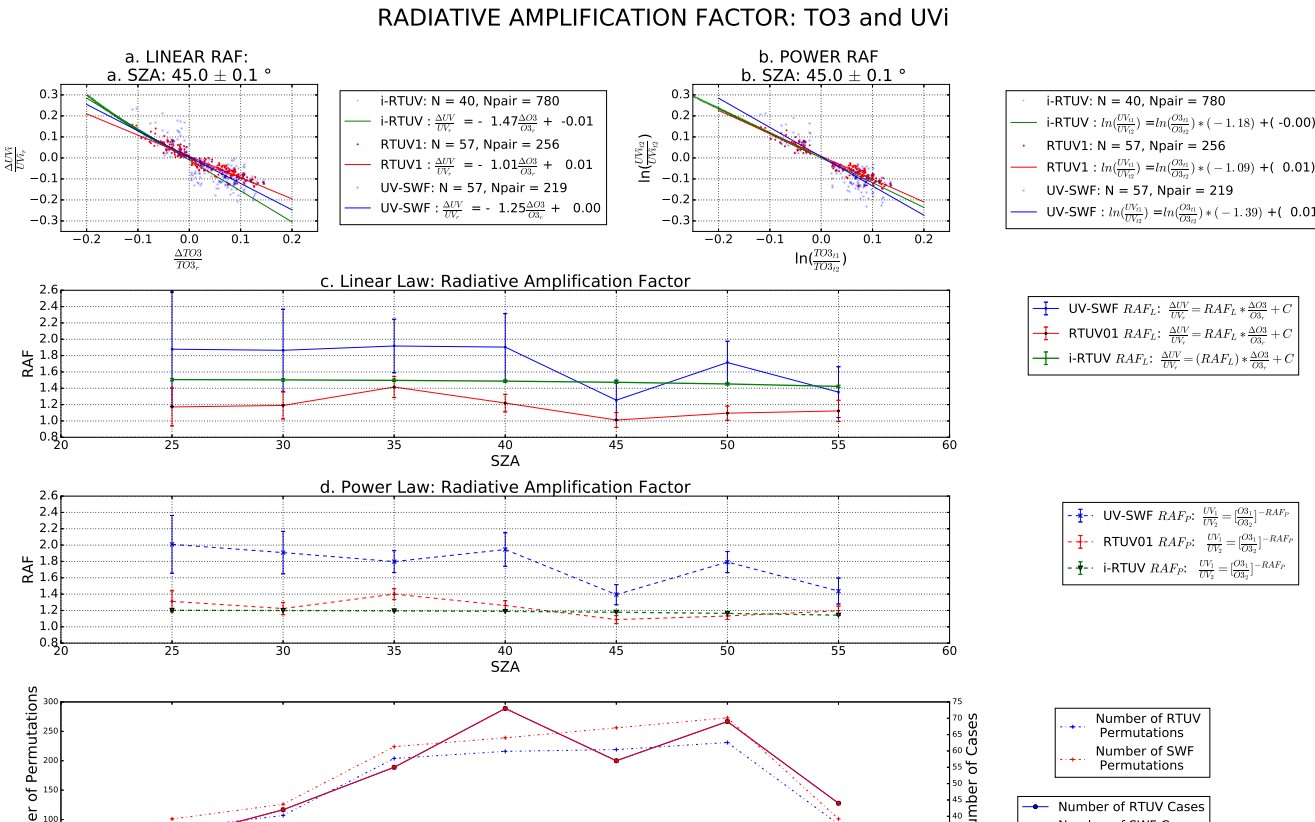

**Figure 8.** Radiative Amplification Factor.

a) $\Delta UV/UV$ against $\Delta TO3/TO3$ for UV-SWF (blue crosses), RTUV01 (red crosses) and i-RTUV (green crosses). Linear fitted functions are in respective coloured curves of corresponding colours. b) Same as 8a but for a power-law fit between $UV/UV_r$ and $O3/O3_r$. c,d) Linear and power RAF deduced from the previous fit for a varying SZA with 2-sigma dispersion bar. f) Number of permutations (Npair) and number of cases (N) available against SZA.



**Figure 9.** Comparisons of RTUV modelling cases with UVI-SWF observations. a) UVI RTUV against UVI SWF. Linear regression in black line, in blue crosses all RTUV data against UVI SWF, and in red crosses, RTUV data modelled with an AOT lower than or equal to the daily mean. b) Mean relative difference for a varying SZA with dispersion bar ± 1 σ (standard deviation)





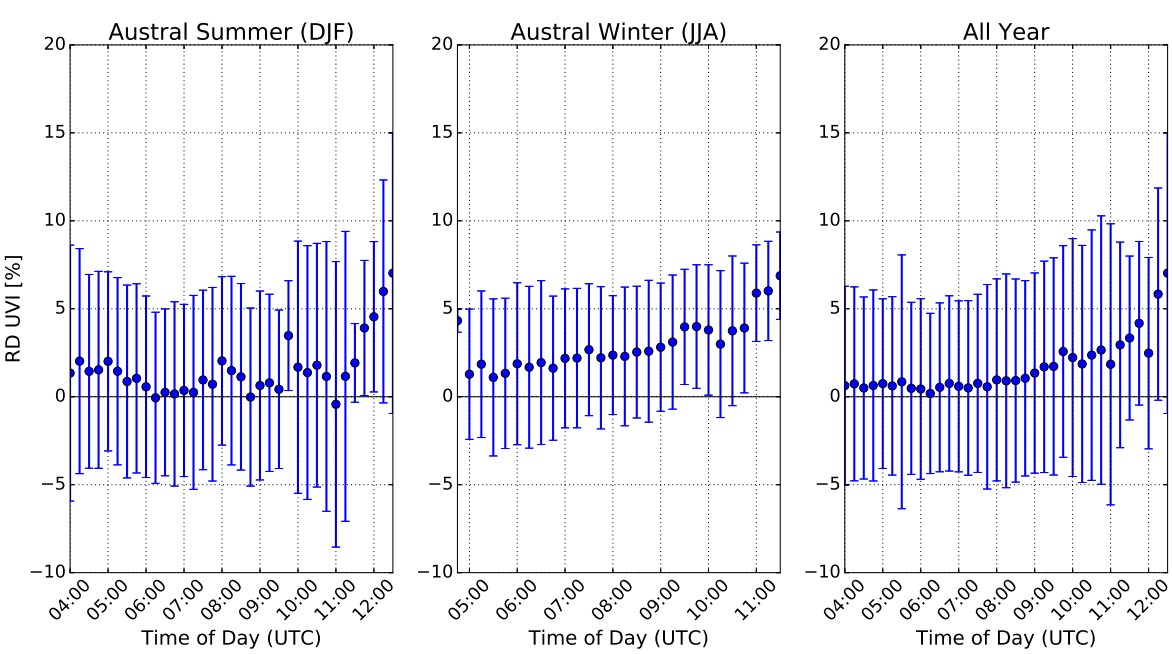

**Figure 10.** Diurnal Cycle of UVI Relative Difference between RTUV01 and UV-SWF for Austral Winter, Summer, and All Year. UVI relative differences every 15 minutes are in blue dots with dispersion bar ( $\pm 1\,\sigma$ (standard deviation)).





**Figure 11.** UVI Diurnal Cycle, Seasonal Climatology. Mean diurnal UVI on the top row and max diurnal UVI on the bottom row. Austral summer on the left side and austral winter on the right. RTUV 06 in blue, UVI-SWF in red, UVI-SYNOP in purple and Uvi-all in green.



**Table 1.** Datasets.

| Data | Instrument | Location | Resolution | Contact(P.I) | Affiliation |
|---|---|---|---|---|---|
| UV index (UV Spectrum integrated following Mc Kinlay and Diffey (1987)) | Bentham DM300 | Saint-Denis Réunion Island University | dt = 15 min dw = 0.5 nm [280-450nm] | C. Brogniez | LOA[1] (Lille-France) |
| Aerosol Optical Thickness at 340nm(AOT) Single Scattering Albedo at 438nm (SSA) Ångström Exponent at 340-440nm ($\alpha$) | Cimel Sunphotometer | Saint-Denis Réunion Island University | Daily Mean | P. Goloub | LOA[1] (Lille-France) |
| Cloud | Observer Report | Saint-Denis Gillot | dt = 1 hour | F. Bonnardot | MF[2] (Saint-Denis) |
| Global and Diffuse Total Irradiance | SPN1 Shaded Pyranometer | Saint-Denis Réunion Island University | dt = 1 min | B. Morel P. Jeanty | LE2P[3] (Saint-Denis) |
| Total Ozone Column ($TO_3$) | SAOZ | Saint-Denis Réunion Island University | Daily | A. Pazmino T.Portafaix | LATMOS (Paris-France) LACy[5] (Saint-Denis) |
| | SBUV2 | Satellite | Daily Overpass | Richard McPeters | NASA |
| | OMI-DOAS | Satellite | Daily Overpass | P. Veefkind | KNMI (Netherland) |
| Total Nitrogen Dioxide Column ($TNO_2$) | SAOZ | Saint-Denis Réunion Island University | Daily | A. Pazmino T.Portafaix | LATMOS (Paris-France) LACy[5] (Saint-Denis) |
| Ozone and Temperature Profil | Ozone sonde | Saint-Denis Réunion Island University | Weekly | F. Posny francoise.posny@univ-reunion.fr | LACy[5] (Saint-Denis) Réunion Island University |
| | MLS | Satellite | Daily Overpass | L. Froidevaux | NASA-JPL |

[a]LOA: Laboratoire d'Optique Atmosphérique de Lille
[b]MF: Météo France
[c]LE2P: Laboratoire d'Energétique, d'Electronique et Procédé
[d]LATMOS: Laboratoire Atmosphères, Milieux, Observations Spatiales
[e]LACy: Laboratoire de l'Atmosphère et des Cyclones





**Table 2.** Base line configuration of the TUV model.

| Parameter | |
|---|---|
| Period | 2009-2016 |
| Latitude | -20.90 |
| Longitude | 55.50 |
| Temporal resolution (dt) | 15 min |
| Vertical scale | 0-80 km |
| Vertical resolution (dz) | 1km |
| Wavelength | 280-450nm |
| Wavelength resolution (dw) | 0.5 nm |
| Albedo | (Koelemeijer et al., 2003) |
| $TNO_2$ | SAOZ |
| $TSO_2$ | 0 |
| Cloud Fraction | 0 |
| $O_3P$ | McPeters and Labow (2012) |
| $TP$ | McPeters and Labow (2012) |
| SSA | 0.95 |
| Aerosol Profile | Elterman (1968) |





**Table 3.** Case configurations

| Case | $TO_3$ | Aerosols | ETS | $O_3$ XS | ESD |
|---|---|---|---|---|---|
| RTUV01 | SAOZ | Cimel Daily | Dobber et al. (2008) | SER | f(t) |
| RTUV02 | OMI | Cimel Daily | Dobber et al. (2008) | SER | f(t) |
| RTUV03 | SBUV | Cimel Daily | Dobber et al. (2008) | SER | f(t) |
| RTUV04 | SAOZ | Cimel Monthly Clim | Dobber et al. (2008) | SER | f(t) |
| RTUV05 | SAOZ | Cimel Daily | Chance and Kurucz (2010) | SER | f(t) |
| RTUV06 | SAOZ | Cimel Daily | Dobber et al. (2008) | BDM | f(t) |
| i-RTUV | 250-345DU | AOT: 0.05 SSA: 0.92 $\alpha$: 0.90 | Dobber et al. (2008) or Chance and Kurucz (2010) | SER or BDM | Constant esfact=1 SZA=[0,60]° |



**Table 4.** RTUV Cases against Clear Sky UVi Observations

| Parameter | RTUV01 | RTUV02 | RTUV03 | RTUV04 | RTUV05 | RTUV06 |
|---|---|---|---|---|---|---|
| Mean Relative Difference [%] | 0.94 | 0.54 | 0.43 | 1.29 | 4.18 | 4.38 |
| Standard Deviation | 5.60 | 5.86 | 5.83 | 5.32 | 5.77 | 5.78 |
| Median of the Mean RD | 1.07 | 0.70 | 0.44 | 0.73 | 4.31 | 4.51 |



*Acknowledgements.* The authors acknowledge the Région Réunion, CNRS and Université de la Réunion for support and contribution within the research infrastructure OPAR (Observatoire de Physique de l'Atmosphère à la Réunion). OPAR is presently funded by CNRS (INSU) and Université de la Réunion and managed by OSU-R (Observatoires des Sciences de l'Univers à la Réunion, UMS 3365). The authors acknowledge Photon Aeronet and NDACC network for the aerosols and ozone data.



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
