# Peer review of "Ultraviolet Radiation modelling from ground based and satellite measurements at Reunion Island, Southern Tropics"

_Atmospheric Chemistry and Physics, 2017_

## Referee Comment (RC1) · Anonymous Referee #1 · 16 Aug 2017

The manuscript presents a comparison of spectrally measured UV-index under clear sky conditions with model calculations for a tropical site over the period 2009 to 2016. First a sensitivity study is reported, where the effect of different sources for local ozone column, of two different extraterrestrial spectra and of two different ozone cross-sections on the calculated UV-index are discussed. These are not really new findings (as stated in 4.2.5, ln. 18), because very similar studies have been carried out already so far, but here they are specifically for the atmospheric conditions of a tropical station. Then in section 5 the model calculations are 'validated' against the observations. In general I think this is an interesting approach, as measurements under such conditions are very rare, and they complete and improve our understanding of UV-levels

at the Earth's surface. Therefore I think the manuscript is worthwhile to be published. However, I think that some specific points should be considered prior to publication:

The abstract could be shorter and more concise, there are some sentences appropriate for the introduction but not for the abstract.

ad 2. Dataset: a bit more information could be given about the calibration of the spectroradiometer (traceability, frequency, . . .).

ad 5.1 Radiative Amplification Factor (RAF):

- There are not 2 different definitions for the RAF. The linear relationship is simply the derivation of the power law. Therefore it is valid only for small variations of ozone and it becomes more and more erroneous for relative variations greater than 5%. The data presented in the manuscript are in the order of up to about 15%, but the systematic deviation between the linear relation and the correct power law is not obvious due to the scatter of the data points.

- The RAF describes the sensitivity of UV to ozone variations, while all other influencing parameters should be constant. Therefore it does not make sense to calculate the RAF including the varying earth-sun-distance (ESD), because the RAF should be the same if ozone is 300 DU in January or July, whereas the UV is higher in January. This means, not the model calculation should include the varying ESD, but the measurements should be converted to a constant ESD. This is of specific importance, when the number of measurements is not constant over the months of the year, because then it will produce a systematic error. Similarly, also the aerosol amount should be constant in the model calculation, whereas in the measurements its variation will produce a significant scatter in the analysis. So the usage of model RTUV01 is not suited to determine the RAF, it only can be used to compare the measured results with modelling (but this is anyway better done by direct comparison of the UV-index derived from measurements and from modelling, as discussed in the following paragraphs).

- what data are used for the calculation of RAF with the ideal model case i-RTUV concerning extraterrestrial spectrum and ozone cross section – in Tab. 3 there are in both cases two different options mentioned with 'or'.

- the calculated RAF with the ideal model case i-RTUV for the power relation (1.2 at SZA=25°) is in very good agreement with the value of 1.25 from Herman [2010] at low SZA, so the statement in ln. 11 ('the higher RAF values found here . . .') is not valid. Furthermore, the argument that the 'lower value of ozone' (ln. 12) is responsible for any difference cannot be true, as the RAF is valid for the whole range of ozone values due to its definition.

ad 5.2 Validation against observed clear-sky UVI: for a validation of different results of model calculations against measurements the significance of the comparison between model and observation should be stated. In this case, the uncertainty of the measurements is +-5% (I guess this holds for a coverage factor of 1). Therefore a mean relative difference between model and measurement in the range 0.4% to 1.3% (Table 4) cannot be significant. Only mean relative differences greater 4% might be significant on a certain level.

Quite many technical corrections: (unfortunately, in my copy of the manuscript the page numbers are missing - this makes the commenting more laborious)

ad 1st page of introduction, ln. 20: not 'during winter' but 'during summer'

ad 2nd page of introduction, ln.18: 'tries' instead of 'tires'

ad 3. Clear-sky filtering, ln. 35: 'with 15 minute intervals' instead of 'with at 15 minute intervals'

ad 4.1, ln. 27: 'by Lacagnina' instead of 'by to Lacagnina'

ad 4.1, ln. 30/31: unclear formulation

ad 4.2.1, ln. 17: 'crossing the atmosphere is longer' instead of 'crossing the atmo-

sphere travelled is longer'

ad 4.2.1, ln. 18: 'processes' instead of 'process'

ad 4.2.4, ln. 32: 'aerosol measurements' instead of 'aerosols measurements'

ad 1st page of 5.1, ln. 40: not a linear relation between UVI and TO3, but between delta(UVI)/UVI and delta(TO3)/TO3

ad 2nd page of 5.1, ln. 17: 'RTUV01' instead of 'RTUV04'

ad 2nd page of 5.1, ln. 18: 'to see how' instead of 'to seehow'

ad 2nd page of 5.1, ln. 31: 'RTUV01' instead of 'RTUV04'

ad 1st page of 5.2, ln. 11: this sentence is almost a repetition of ln. 5

ad 1st page of 5.2, ln. 13: 'SAOZ' instaed of 'SBUV'

ad 1st page of 5.2, ln. 40-43: this paragraph is a duplication of ln. 1 and 2 and does not fit here

ad 1st page of 5.2, ln. 42: duplicate 'on on'

ad 2nd page of 5.2, ln. 11: 'which increased' instead of 'with increased'

ad 1st page of 6, ln. 47: for which 'higher SZA' the given numbers are derived?

ad 2nd page of 6, ln. 11: duplicate 'to to'

Table 4, last line: 'Median of the RD' instead of 'Median of the Mean RD'

Figure 8: the figures 8a and 8b are by far too small to see the different data points

Figure8: the legend says '8f', but this is not shown in the figure itself. May be '8e' would be appropriate (but not labelled). Anyway, this last part of Fig. 8 could be skipped.

Figure11: in the Figure 'RTUV01' is mentioned, in the legend below the figure it says 'RTUV06'

Overall I think the manuscript is worthwhile to be published in ACP after revision.

[Figure]

---

## Referee Comment (RC2) · Anonymous Referee #2 · 18 Aug 2017

**Review of the manuscript with title: "Ultraviolet Radiation modeling from ground based and satellite measurements at Reunion Island, Southern Tropics" from Lamy et al.**

I suggest that the following comments have to be addressed before the manuscript is suitable for publication to the ACP.

**General comments**

There are still several syntactical, linguistic and editorial errors in the manuscript, especially in the abstract and paragraph 4.2.4. Thus, I suggest that the authors should try harder to improve it. In the following there are indicative recommendations for a number of corrections, but there are more that have to be done and additional work is necessary.

In the present study a standard SSA of 0.95 was used. Though, there are studies suggesting that the SSA in the UV may range between much lower values (e.g. ~0.6) and values close to unity. Although for very low values of AOT (such as those that are usual at Reunion Island) changing the SSA would not induce important changes in the model's output, I believe that it would be useful to quantify the possible errors/uncertainties due to the use of a standard SSA.

The differences between the simulated UVI for different ETS are surprisingly large. I believe that the dependence from ozone and SZA denotes very large differences between the UV-B wavelengths of the spectra from Dobber and Chance & Kurucz. Are there any other differences between these two spectra (e.g. different spectral resolution/analysis) which could induce such large differences in the simulated UVI?

**Specific comments**

Abstract

The text of the abstract should be rearranged so that its meaning is clearer. For example:

- The first two paragraphs of the introduction could be rearranged so that it is easier for the reader to follow them. The first paragraph should answer to the question: "why studying SUR is important?" while the second paragraph should answer to the question: "why studying SUR over the tropics – and in particular in the Reunion Island – is important?"
- After the first two paragraphs, there should be a general description of what has been done in this study, i.e. move the text of P2, L1-3 there.
- The discussion for the cloud filtering is also divided in two paragraphs (5 and 6). I suggest making this discussion in a single paragraph (i.e. move the relative information from the last paragraph since it is not one of the main findings of the study).

Some suggestions for technical corrections are the following:

P1, L4: Define SUR here instead of P2, L20

P1, L18: Delete "radiation"

P2, L4: "SUR was" instead of "ultraviolet radiations were"

P2, L5: "was based" instead of "based" and "while the second was based on applying" instead of "the second applying"

P2, L10: "were derived" instead of "came"

P2, L11: "using" instead of "with respect to"

1.  Introduction

P2, L20: "However, large" instead of "Large"

P2, L21: Delete "As"

P2, L28: "depends" instead of "depending"

P3, L8: Do you mean "by absorbing and scattering processes in the atmosphere" instead of "by the atmosphere and scattering processes"?

P3, L23: "and" instead of "they".

P3, L23: What caused the reduction of 15.2%?

P3, L24: "usually reduce" instead of "can reduce"

P3, L27: "role" instead of "part"

P3, L34: "integral" instead of "integration"

P4, L11: "projections" instead of "projection"

P4, L32: "parameters" instead of "parameter"

2.  Datasets

P5, L20: rephrase

P5, L27-28: which one is the first paper and which one is the second paper?

P6, L1-8: I think that this information is not related with the present work and I suggest removing it.

P6, L12: The cloud observations are performed at a distance of 10 km from the location of the measurements, but where (i.e. east, west, …)?

3. Clear-sky filtering

For consistency I suggest using either "clear-sky" or "clear sky" throughout the entire manuscript.

P6, L19: "observations" instead of "observation"

P7, L6: "relative to" instead of "higher than"

P7, L8: "At around" instead of "Around"

P7, L13: Delete "with"

4. UV modeling

P7, L28: I suggest explaining in short why the used approximation is more accurate.

P8, L11: "performed" instead of "taken"

P10, L20-P11, L13: This paragraph is very badly written and confusing. I suggest rewriting it more carefully.

5. Model validation

P11, L15: Declare that you compare the observations with the model output.

P11, L17: I suppose that you mean the sensitivity of the model output on TO3. You should make it clear here.

P12, L6: what means "very clear sky"?

Figure 8: In my opinion, the equations are not necessary in the legends of figures 8(a) – 8(d) since there is an explanation of what is $RAF_P$ and $RAF_L$. In the legend of figures 8(a) and 8(b), the value of RAF would be enough.

P12, L18: "This range of SZAs is" instead of "These ranges of SZA are", "measurements" instead of "measurement", "the annual" instead of "annual", "lower" instead of "low"

P12, L20: "in figure" instead of "on figure". This is applicable to the entire manuscript.

P13, L14: Writing that the agreement between the measured and modeled UVIs is the best when the SBUV dataset is used as input would be more accurate than writing that the results RTUV03 is the best. The fact that the agreement is optimal does not necessarily mean that RTUV03 is the best.

Figure 9: The title of the x-axis is below each sub-figure of fig. 9a, while at fig 9b it is only below the last sub-figure. For consistency I suggest removing the x-axis title from the five upper panels of fig 9a.

---

## Short Comment (SC1) · 17 Sep 2017

Response and update manuscript attached in the zip file.

Please also note the supplement to this comment:
https://www.atmos-chem-phys-discuss.net/acp-2017-571/acp-2017-571-SC1-supplement.zip

———————————————————

---

## Short Comment (SC2) · 17 Sep 2017

Response and updated manuscript attached as a supplement.

Please also note the supplement to this comment:
https://www.atmos-chem-phys-discuss.net/acp-2017-571/acp-2017-571-SC2-supplement.zip
* * *

---

## Author Comment (AC1) · 30 Oct 2017

The authors would like to thank the editor and the reviewers for their time and comments. We have addressed both referee's comments in the discussion. The corresponding changes are detailed in the short comment 1(Response to referee 1) and 2 (Response to referee 2) in the discussion.

From these changes and the latest co-authors review, a final version was written and is attached as a supplement to this comment.

The latest changes include :

[Figure]

- Corrections which are detailed in the short comment 1 and 2. (Response to referee 1 and 2)

- Additional syntax and grammar correction.

- Figures have been tweaked to be easier to read (line width or font size have been increased).

- References have been added (P2, L15 and P3, L19).

- Information about the threshold, the number of total and filtered measurements have been added (P3, L15).

- Figure 7: Only AOT at 340nm, SSA at 440nm and Ångström exponent at 340-440nm have been kept.

Please also note the supplement to this comment:
https://www.atmos-chem-phys-discuss.net/acp-2017-571/acp-2017-571-AC1-supplement.pdf

---

## Author Response (AR1)

**Finale Response**

Kévin Lamy, Thierry Portafaix, Colette Brogniez, Sophie Godin-Beekman,
Hassan Bencherif , Béatrice Morel, Andrea Pazmino, Jean Marc Metzger,
Frédérique Auriol, Christine Deroo, Valentin Duflot, Philippe Goloub, Charles N. Long

October 30, 2017

The authors would like to thank the editor and the reviewers for their time and comments. We have addressed both referees comments in the discussion. The corresponding changes are detailed in the short comment 1 (in response to referee 1) and 2 (response to referee 2) in the discussion.

From these changes and the latest co-authors review, a final version was written.

The latest changes realised after the short comments 1 and 2 include :

- Corrections which are detailed in the short comment 1 and 2. (Response to referee 1 and 2)

- Additional syntax and grammar correction.

- Figures have been tweaked to be easier to read (line width or font size have beenincreased).

- References have been added (P2, L15 and P3, L19).

- Information about the threshold, the number of total and filtered measurements have been added (P3, L15).

- Figure 7: Only AOT at 340nm, SSA at 440nm and ngstrm exponent at 340-440nm have been kept.

The following items are included in this pdf:

- P1-5: Response to referee 1.

- P6-9: Response to referee 2.

- P10-46: Finale version of the manuscript.

**Response to the Referee's Comments**

Kévin LAMY

September 17, 2017

**1 General comment**

> The manuscript presents a comparison of spectrally measured UV-index under clear sky conditions with model calculations for a tropical site over the period 2009 to 2016. First a sensitivity study is reported, where the effect of different sources for local ozone column, of two different extraterrestrial spectra and of two different ozone cross sections on the calculated UV-index are discussed. These are not really new findings (as stated in 4.2.5, ln. 18), because very similar studies have been carried out already so far, but here they are specifically for the atmospheric conditions of a tropical station. Then in section 5 the model calculations are 'validated' against the observations.
>
> In general I think this is an interesting approach, as measurements under such conditions are very rare, and they complete and improve our understanding of UV-levels at the Earth's surface. Therefore I think the manuscript is worthwhile to be published.

We would like to thank the referee for his thorough review. The comments on the radiative amplification factor and the structure of the article have been very beneficial. We hope that the newer version has improved the paper.

**2 Specific comments**

> The abstract could be shorter and more concise, there are some sentences appropriate for the introduction but not for the abstract.

The abstract is now shorter and was restructured to be clearer.

> ad 2. Dataset: a bit more information could be given about the calibration of the spec- troradiometer (traceability, frequency, . . .).

Specific information have been added.

> ad 5.1 Radiative Amplification Factor (RAF):
> - There are not 2 different definitions for the RAF. The linear relationship is simply the derivation of the power law. Therefore it is valid only for small variations of ozone and it becomes more and more erroneous for relative variations greater than 5%. The data presented in the manuscript are in the order of up to about 15%, but the systematic deviation between the linear relation and the correct power law is not obvious due to the scatter of the data points.

The referee is right about the relationship between the power law and the linear law. It has been taken into account, we used now only relative variations lower than 10%. We tried 5% at first but not enough data remained.

> - The RAF describes the sensitivity of UV to ozone variations, while all other influencing parameters should be constant. Therefore it does not make sense to calculate the RAF including the varying earth-sun-distance (ESD), because the RAF should be the same if ozone is 300 DU in January or July, whereas the UV is higher in January. This means, not the model calculation should include the varying ESD, but the measurements should be converted to a constant ESD. This is of specific importance, when the number of measurements is not constant over the months of the year, because then it will produce a systematic error. Similarly, also the aerosol amount should be constant in the model calculation, whereas in the measurements its variation will pro- duce a significant scatter in the analysis. So the usage of model RTUV01 is not suited to determine the RAF, it only can be used to compare the measured results with mod- elling (but this is anyway better done by direct comparison of the UV-index derived from measurements and from modelling, as discussed in the following paragraphs).

The referee is correct. We also changed this part, now there is a new simulation RTUV07 which is similar to RTUV01 but with constant aerosols, total no2 column , etc... Only total ozone column, sza and ozone and temperature profil are not constant. It allowed to us to have an idea of the impact of ozone and temperature profil on the RAF. The observations were converted to a constant ESD by multiplying the UVI by $1/esfact$.

> - what data are used for the calculation of RAF with the ideal model case i-RTUV concerning extraterrestrial spectrum and ozone cross section  in Tab. 3 there are in both cases two different options mentioned with 'or'.

Serdyuchenko et al. (2014) ozone cross section and Dobber et al. (2008) extraterrestrial spectrum is used in i-RTUV for the RAF calculation. Clarification have been added in the manuscript.

> - the calculated RAF with the ideal model case i-RTUV for the power relation (1.2 at SZA=25  ) is in very good agreement with the value of 1.25 from Herman [2010] at low SZA, so the statement in ln. 11 ('the higher RAF values found here . . .') is not valid. Furthermore, the argument that the 'lower value of ozone' (ln. 12) is responsible for any difference cannot be true, as the RAF is valid for the whole range of ozone values due to its definition.

The referee is right. The statement have been removed.

> ad 5.2 Validation against observed clear-sky UVI: for a validation of different results of model calculations against measurements the significance of the comparison between model and observation should be stated. In this case, the uncertainty of the measurements is +-5% (I guess this holds for a coverage factor of 1). Therefore a mean relative difference between model and measurement in the range 0.4% to 1.3% (Table 4) cannot be significant. Only mean relative differences greater 4% might be significant on a certain level.

We have added a statement about the significance of the comparison.

**3   Technical corrections**

The following errors have been corrected. Since the structure and have changed, corresponding pages and lines in the newer version are written below.

> 1. ad 1st page of introduction, ln. 20: not 'during winter' but 'during summer'

Corrected. P2,L22

> ad 2nd page of introduction, ln.18: 'tries' instead of 'tires'

Corrected. P3,L34

> ad 3. Clear-sky filtering, ln. 35: 'with 15 minute intervals' instead of 'with at 15 minute intervals'

Corrected. P7,L4

> ad 4.1, ln. 27: 'by Lacagnina' instead of 'by to Lacagnina'

Corrected. P8,L13

> ad 4.1, ln. 30/31: unclear formulation

Removed in the newer version.

> ad 4.2.1, ln. 17: 'crossing the atmosphere is longer' instead of 'crossing the atmosphere travelled is longer'

Corrected. P9,L11

> ad 4.2.1, ln. 18: 'processes' instead of 'process'

Corrected. P9,L11

> ad 4.2.4, ln. 32: 'aerosol measurements' instead of 'aerosols measurements'

Removed in the newer version.

> ad 1st page of 5.1, ln. 40: not a linear relation between UVI and TO3, but between delta(UVI)/UVI and delta(TO3)/TO3

Removed in the newer version.

> ad 2nd page of 5.1, ln. 17: 'RTUV01' instead of 'RTUV04'

Corrected.

> ad 2nd page of 5.1, ln. 18: 'to see how' instead of 'to seehow'

Corrected. P12,L15

> ad 2nd page of 5.1, ln. 31: 'RTUV01' instead of 'RTUV04'

Corrected.

> ad 1st page of 5.2, ln. 11: this sentence is almost a repetition of ln. 5

Removed in the newer version.

> ad 1st page of 5.2, ln. 13: 'SAOZ' instaed of 'SBUV'

Corrected. P13,L22

> ad 1st page of 5.2, ln. 40-43: this paragraph is a duplication of ln. 1 and 2 and does not fit here

Removed in the newer version.

> ad 1st page of 5.2, ln. 42: duplicate 'on on'

Corrected.

> ad 2nd page of 5.2, ln. 11: 'which increased' instead of 'with increased'

Corrected. P14,L26

> ad 1st page of 6, ln. 47: for which 'higher SZA' the given numbers are derived?

Corrected. P15,L26

> ad 2nd page of 6, ln. 11: duplicate 'to to'

Corrected. P16,L10

> Table 4, last line: 'Median of the RD' instead of 'Median of the Mean RD'

Corrected.

> Figure 8: the figures 8a and 8b are by far too small to see the different data points

Corrected.

> Figure8: the legend says '8f', but this is not shown in the figure itself. May be '8e' would be appropriate (but not labelled). Anyway, this last part of Fig. 8 could be skipped.

Corrected.

> Figure11: in the Figure 'RTUV01' is mentioned, in the legend below the figure it says 'RTUV06'

Corrected.

**Response to the Referee's Comments**

**Kévin LAMY**

**September 17, 2017**

I would like to thank the referee for his second review. We hope that the newer version has improved the paper

**1 General comment**

There are still several syntactical, linguistic and editorial errors in the manuscript, especially in the abstract and paragraph 4.2.4. Thus, I suggest that the authors should try harder to improve it.In the following there are indicative recommendations for a number of corrections, but there are more that have to be done and additional work is necessary.

In the present study a standard SSA of 0.95 was used. Though, there are studies suggesting that the SSA in the UV may range between much lower values (e.g. 0.6) and values close to unity. Although for very low values of AOT (such as those that are usual at Reunion Island) changing the SSA would not induce important changes in the models output, I believe that it would be useful to quantify the possible errors/uncertainties due to the use of a standard SSA.

The differences between the simulated UVI for different ETS are surprisingly large. I believe that the dependence from ozone and SZA denotes very large differences betweenthe UV-B wavelengths of the spectra from Dobber and Chance & Kurucz. Are there any other differences between these two spectra (e.g. different spectral resolution/analysis) which could induce such large differences in the simulated UVI?

Regarding the general comment, we tried to correct the remaining syntactical, linguistic and editorial errors. We greatly appreciate the effort the referee did to point them out in the specific comments.

We agree with the reviewer's comment about SSA. It would be useful to quantify errors due to a constant SSA but we feel like it would be more suited for a future study dedicated on UVI modelling sensitivity to aerosols.

The referee is right, most of the difference between Dobber and Chance & Kurucz spectra are in the UV-B range. Nonetheless, as input for TUV model both spectra have a 0.01 nm resolution.

**2 Specific comments**

The following errors have been corrected.

**2.1 Abstract**

The text of the abstract should be rearranged so that its meaning is clearer. For example: The first two paragraphs of the introduction could be rearranged so that it is easier for the reader to follow them. The first paragraph should answer to the question: why studying SUR is important? while the second paragraph should answer to the question: why studying SUR over the tropics - and in particular in the Reunion Island is important? After the first two paragraphs, there should be a general description of what has been done in this study, i.e. move the text of P2, L1-3 there. The discussion for the cloud filtering is also divided in two paragraphs (5 and 6). I suggest making this discussion in a single paragraph (i.e. move the relative information from the last paragraph since it is not one of the main findings of the study).

Following the reviewer suggestions the structure of the abstract has been reworked. It should be now easier to read.

1. P1,L4: Define SUR here instead of P2, L20

Corrected. P1, L1

2. P1, L18: Delete radiation

Corrected. P1, L15

3. P2, L4: SUR was instead of ultraviolet radiations were

Corrected. P2, L1

4. P2, L5: was based instead of based and while the second was based on applying instead of the second applying

Corrected. P2, L2

5. P2,L10: "were derived" instead of "came"

Corrected. P2, L9

6. P2,L11: "using" instead of "with respect to"

Corrected.

**2.2 Introduction**

7. P2, L20: However, large instead of Large

Corrected. P2, L15

8. P2, L21: Delete As

Corrected. P2, L16

9. P2, L28: depends instead of depending

Corrected. P2, L24

10. P3, L8: Do you mean by absorbing and scattering processes in the atmosphere instead of by the atmosphere and scattering processes?

Corrected. P3, L3

11. P3, L23: and instead of they.

Corrected. P3, L18

12. P3, L23: What caused the reduction of 15.2%?

Corrected. P3, L18

13. P3, L24: usually reduce instead of can reduce

Corrected. P3, L19

14. P3, L27: role instead of part

Corrected. P3, L22

15. P3, L34: integral instead of integration

Corrected. P3, L29

16. P4, L11: projections instead of projection

Corrected. P4, L6

17. P4, L32: parameters instead of parameter

Corrected. P4, L27

**2.3 Dataset**

18. P5, L20: rephrase

Corrected. P5, L18

19. P5, L27-28: which one is the first paper and which one is the second paper?

Clarification has been added. P5, L25

20. P6, L1-8: I think that this information is not related with the present work and I suggest removing it.

We removed the information about the ASCO activity but kept Orphal et al. (2016) result in a more concise form. P5, L27

21. P6, L12: The cloud observations are performed at a distance of 10 km from the location of the measurements, but where (i.e. east, west, ...)?

Corrected in the manuscript, the cloud observations are performed 10 km north of the measurements. P6, L4

**2.4 Clear-sky filtering**

22. For consistency I suggest using either clear-sky or clear sky throughout the entire manuscript.

Corrected. "Clear-sky" is now used throughout the entire manuscript.

23. P6, L19: observations instead of observation

Corrected. P6, L11

24. P7, L6: relative to instead of higher than

Corrected. P6, L27

25. P7, L8: At around instead of Around

Corrected. P6, L30

26. P7, L13: Delete with

Corrected. P7, L3

**2.5  UV modeling**

27.  P7, L28: I suggest explaining in short why the used approximation is more accurate.

Corrected. P7, L19

28. P8, L11: performed instead of taken

Corrected. P8, L9

29. P10, L20-P11, L13: This paragraph is very badly written and confusing. I suggest rewriting it more carefully.

Corrected. P10, L21 - P5, L14

**2.6  Model Validation**

30. P11, L15: Declare that you compare the observations with the model output.

Corrected. P11, L14

31. P11, L17: I suppose that you mean the sensitivity of the model output on TO3. You should make it clear here.

Corrected. P11, L16

32. P12, L6: what means very clear sky?

Following on the first referee's comment, the entire statement here was deleted.

33. Figure 8: In my opinion, the equations are not necessary in the legends of figures 8(a)  8(d) since there is an explanation of what is RAF P and RAF L . In the legend of figures 8(a) and 8(b), the value of RAF would be enough.

Corrected.

34. P12, L18: This range of SZAs is instead of These ranges of SZA are, measurements instead of measurement, the annual instead of annual, lower instead of low

Corrected. P12, L20

35.  P12, L20: in figure instead of on figure.  This is applicable to the entire manuscript.

Corrected.

36. P13, L14: Writing that the agreement between the measured and modeled UVIs is the best when the SBUV dataset is used as input would be more accurate than writing that the results RTUV03 is the best. The fact that the agreement is optimal does not necessarily mean that RTUV03 is the best.

Corrected. P13, L20

37. Figure 9: The title of the x-axis is below each sub-figure of fig. 9a, while at fig 9b it is only below the last sub-figure. For consistency I suggest removing the x-axis title from the five upper panels of fig 9a.

Corrected. P25

[revised manuscript text omitted]